# Isolation of *Lactococcus lactis* from Whole Crop Rice and Determining Its Probiotic and Antimicrobial Properties towards Gastrointestinal Associated Bacteria

**DOI:** 10.3390/microorganisms9122513

**Published:** 2021-12-03

**Authors:** Ilavenil Soundharrajan, Yong Hee Yoon, Karnan Muthusamy, Jeong-Sung Jung, Hyun Jeong Lee, Ouk-Kyu Han, Ki Choon Choi

**Affiliations:** 1Grassland and Forages Division, National Institute of Animal Science, Rural Development Administration, Cheonan 31000, Korea; ilavenil@korea.kr (I.S.); karnantm111@gmail.com (K.M.); jjs3873@korea.kr (J.-S.J.); 2Jungnong Bio Inc., 40-4, Chumdan 4-Ro, Jeongeup 56212, Korea; jnbioyoon@gmail.com; 3Jangsu Agriculture Technology Center, Jangsu 55640, Korea; leehj5329@korea.kr; 4Department of Crop Science, Korea National College of Agriculture and Fisheries, Jeonju 54874, Korea; okhan98@korea.kr

**Keywords:** infectious diseases, probiotics, *Lactococcus lactis*, CFS, antagonistic activities

## Abstract

Antimicrobial resistance is an emerging condition that increases the risk of spreading and prolonging infectious diseases globally. Therefore, a new alternative strategy for antibiotics is required urgently to control pathogens spreading. Probiotics are considered as an alternative for antibiotics that inhibit pathogens. In the present study, potent lactic acid bacteria (LAB) were isolated and screened for their probiotic characteristics and antagonistic activity against intestinal pathogens by agar well diffusion, Time and Dose-dependent killing assay, minimum inhibitor, and minimum bactericidal concentration (MIC/MBC), and co-culture methods. The *Lactococcus lactis* RWP-3 and RWP-7 fermented the different carbohydrate substrates and produced different extracellular enzymes. Both isolates showed significant tolerant capability in the gastric, duodenal, and intestinal juices. In addition, RWP-3 and RWP-7 had hydrophobicity and aggregation properties in a time-dependent manner. Furthermore, the cell-free secondary metabolites (CFS) of RWP-3 and RWP-7 showed strong antibacterial activity against *Escherichia coli,*
*Staphylococcus aureus**, Pseudomonas aeruginosa* and *Enterococcus faecalis*. A co-culture study revealed that the RWP-3 and RWP-7 strongly compete with pathogen growths. RWP-3 and RWP-7 showed strong antagonistic activities against tested pathogens with significant probiotic characteristics, suggesting that these strains obtained could be used as an alternative strategy for the antibiotic to control infectious pathogens.

## 1. Introduction

Outbreaks of infectious diseases have been increased steadily during the 30 years. According to an analysis of 10,643 outbreaks were reported in 2014 [1]. Antimicrobial resistance is an emerging issue and it insists to take a greater concentration of antibiotics resulting in negative consequences in the health, environmental and agriculture sectors [2]. In addition, increases in antibiotic resistance lead to increases in the risk of spreading and prolonging infectious diseases. World health organization, 2017 listed the antibiotics required for following pathogenic bacteria which include *Enterococcus* spp., *Staphylococcus aureus*, *Klebsiella pneumoniae*, *Acinetobacter baumannii, Psedomonas aeruginosa*, and *Enterobacter* sp. [3]. In general pathogenic bacteria and their toxins often enter into the human body via food or drinks causing symptoms or illness with a different mechanism. Approximately 50.2% of *S. aureus* and *E. coli* have been found in the normal flora of people; sometimes it has virulent nature, resistant to common antibiotics which cause sepsis and severe infections. In addition, enterococcus strains dominate pathogenesis in the gastrointestinal tract (GIT) are *E. faecalis* and *E. faecium* [4]. The invention of antibiotics is a major step in the medical field to control deathly infections caused by pathogenic bacteria. However, the ineffectiveness of antibiotics against pathogens has been increased as drug-resistant which spreads globally causing it to be more difficult to treat such infections and death [5]. Therefore, new alternative antibiotics are urgently required to control the spread of pathogens. In general, probiotics are represented as a potential alternative for antibiotics to control and prevention of pathogenic bacteria. Strains belonging to lactic acid bacteria (LAB) such as *Lactobacillus* and *Bifidobacterium* are commonly used as a probiotic and starter culture for several fermentation processes [6,7,8]. These bacteria can produce various antimicrobial agents that exert strong antagonistic activity against different pathogenic microbes. The mechanisms underlying the LAB activity against pathogens appear to be multifactorial ways by various metabolites [9,10,11,12]. LAB can prevent the adhesion of pathogens by competing for the binding sites on the intestinal epithelial cells and reducing the colonization, thereby preventing the onset of infections [13,14]. Among LABs, *Lactococcus lactis* is considered a potent probiotic that improves GIT health. The *L. lactis* usually synthesize bacteriocins that compounds fight against the pathogenic microbes. The *Lactococcus* has received the grade of Generally Recognized as Safe (GRAS) status by the Food and Drug and Administration [15]. The *L. lactis* species are synthesized antimicrobial compounds (e.g. Nisin peptide). The compounds are currently approved for commercial purposes in ≥50 countries [16]. The reports are comprehensively studied for the antimicrobial activity of *L. lactis* that inhibits the pathogenic *E. coli, Enterococcus feacalis, P. aeruginosa, S. aureus* and *Bacillus* species [17]. In the present study, we isolated several LABs (*Lactococcus lactis)* and characterized their probiotic potential with antibacterial activity against intestinal and urinary tract infections causing pathogens such *E. coli, S. aureus, P. aeruginosa* and *E. faecalis* by agar-well diffusion, Time and dose-dependent killing assay MIC/MBC and co-culture method.

## 2. Methods and Materials

### 2.1. Isolation and Characterization of LAB

Whole crop rice (WCR) samples were collected from different places in the same land at Grassland and Forage Division, Seonghwan, Cheonan, Korea. LABs were isolated by MRS agar and confirmed their identity by BCP agar [11,18]. The extracellular enzymes and carbohydrates fermentation were determined by API-CH50 and API-ZYM kits, Marcy-I’ Etoile, France, respectively. The 16srRNA sequence of isolates was analyzed (Solgent Pvt Ltd., Daejeon, Korea) and the sequences were used to identify the isolates at the species level by the NCBI blast tool (GenBank Accession Numbers: RWP-3: OL677065 RWP-7: OL677066. The sequences were aligned with the MUSCLE tool. The evolutionary history was explored using the neighbor-joining method [19].

### 2.2. Probiotic Potential of Selected LAB

Different ranges of pH (2.5, 5, 8.0 pH values) in PBS were prepared with 1M HCl or 1M NaOH and autoclaved at 121 °C with a pressure level at 15 psi. The simulated gastrointestinal juices such as gastric juice (PBS with 3 mg/mL pepsin. pH. 2.5) duodenal (PBS with 0.3% bile salts and 0.1% trypsin, pH. 5), and intestinal juice (PBS with 0.1% trypsin, pH.8) were prepared and filtered according to a previously published protocol [18,20]. Twenty-four bacterial cultures were centrifuged by centrifugation at 4000× *g* for 15 min at 4 °C. The collected pellets were washed with PBS and counted by a quantum live-cell staining kit. The pellets were suspended in PBS and the equal numbers of bacterial colonies (1 mL) were loaded into 9 mL of gastric juice and incubated at 37 °C for 3 h. One milliliter of bacterial colonies was transferred after 3h from gastric juice to duodenal juice and incubated in the same condition. Similarly, after 3 h, 1 mL of bacterial colonies were transferred to intestinal juice from duodenal juice and incubated same conditions. The survival of bacterial colonies was determined every one hour. Hydrophobicity features with chloroform and xylene [18,21] and aggregation [18,22] properties of strains were also determined.

### 2.3. Antibacterial Activity Well Diffusion

The RWP-3 and RWP-7 were cultured in MRS broth in a 1000 mL glass bottle containing butyl stoppers with aluminum crimp and incubated in an arbitrary shaker at 32 °C for 48 h. The CFS was prepared by centrifugation at 4000× *g*, 4 °C for 30 min and the supernatant was collected and used for antibacterial activity. The pathogens such as *E. coli, S. aureus, P. aeruginosa* and *E. faecalis* were obtained from KACC, Korea and then cultured in Nutrient broth and incubated at 37 °C for 24 h. Then 10^8^ CFU/mL o from each pathogen were spread on nutrient agar plates and then made well on it by the well cutter. A hundred microliters of cell-free supernatant were then loaded into an appropriate well and incubated at 37 °C for 48 h. The zone of inhibition of CFS was monitored after 24 h to 48 h.

### 2.4. Lyophilization of Cell Free Metabolites (CFS)

The RWP-3 and RWP-7 were cultured in MRS broth in a 1000 mL glass bottle containing butyl stoppers with aluminum crimp and incubated in an arbitrary shaker at 32 °C for 48 h. Prepared CFS by centrifugation at 4000× *g* for 30 min at 4 °C and the supernatant was collected and neutralized with 1N sodium hydroxide and filtered by different sizes of membrane filters (Whatman no.1, Syringe filters 0.45 µm and 0.22 µm). The filtered CFS was lyophilized below −40 °C in less than 50 m Torr pressure (Ilshin Biobase, Gyeonggi-do, Korea).

### 2.5. Time and Dose-Dependent Killing Assay and Minimum Bactericidal (MIC) and Minimum Inhibitory Concentration (MBC)

Cell-free metabolites (CFS) of selected strains were suspended in Nutrient broth at the concentration of 25 mg/mL and then twofold serial dilution was performed with media (25 mg/mL–0.02 mg/mL). A hundred microliters of serially diluted samples from each concentration were transferred into 96 well plates (*n* = 3). Ten microliters of fresh pathogens (*E. coli, S. aureus, P. aeruginosa* and *E. faecalis*) were inoculated into respective wells. Without CFS was considered as control and blank was also maintained for all the concentrations. All plates were incubated at 37 °C. Aliquots of samples were removed from each group at every 12, 24, 36 and 48 h and read at 600 nm. Final optical density values were normalized with respective blanks [18]. The minimum bactericidal and minimum inhibitory concentration of CFS was determined [18,23].

### 2.6. Antagonistic Activity by Co-Culture Method

The antagonistic properties of the selected isolates against different pathogens were determined by the Co-Culture method [24,25] with slight modifications. The LAB and pathogens were cultured in MRS and nutrient broths, respectively and incubated at suitable temperatures for 24 h and then centrifuged at 4000× *g* in 4 °C for 30 min and washed twice with PBS and suspended in the same buffer. Each LAB isolate was co-cultured with each pathogen in 96 plates well-containing MRS: NA broth (1:1 ratio). For control (monoculture), LAB and Pathogen were individually inoculated in MRS: NA broth and the plates were incubated at 37 °C with mild shaking. The sampling was performed at 12, 24 and 36 h. The bacterial colonies were counted in both MRS and NA agar plates and the data were compared with respective monocultures.

### 2.7. Statistical Analysis

Data obtained from experiments were subjected to statistical analysis by SPSS 16 with (one-way ANOVA, multivariate analysis, including post hoc, Duncan and descriptive analysis parameters. The data were represented as mean ± SEM of three replicates. Less than 0.05 levels of *p*-values were considered statistically significant.

## 3. Results

### 3.1. Isolation and Characterization

Preliminarily, several bacteria were isolated from whole crop rice using an MRS agar media and performed for their antimicrobial activity against various pathogenic microbes. The results exhibited that the two isolates (RWP-3 and RWP-7) showed significant antimicrobial activities than the other isolates (data not shown). These isolates were cocci bacilli, Gram-positive, catalase-negative with creamy colonies in the MRS agar, which confirmed that all isolates belong to the LAB group (Appendix A). Further, we screened for their carbohydrates utilization and enzyme secretions properties. These isolates were utilized several carbohydrate substrates (Appendix A). Also, these strains were secreted by various extracellular enzymes (Appendix A). The sequences of LAB were subjected to the NCBI blast tool, suggesting that the selected strains belonged to *Lactococcus lactis* (>99%). The phylogenetic trees RWP-3 and RWP-6 were constructed from an evolutionary distance by the neighbor-joining method (Appendix A).

### 3.2. RWP-3 and RWP-7 in Simulated Gastrointestinal Tract (GIT)

The probiotic bacteria must be survived in harsh conditions of GIT including low pH with pepsin (Gastric juice, pH 2.5), bile salts with trypsin (duodenal Juice, pH 5) and higher pH with trypsin (Intestinal Juice, pH 8). It is an essential criteria for the selection of potential probiotics. Survival abilities in Gastric juice after 3hrs incubation for RP3-3 and RP37 were 34.22% and 32.93% respectively. The survival ability of RP3-3 (68.79%) and RP37 (62.41%) were higher in duodenal Juice compared to Gastric juice. RP3-3 and RP37 in Intestinal Juice showed higher survival rates (80.90% vs. 87.97%, respectively) compared to Gastric and duodenal juices (Figure 1A).

### 3.3. Hydrophobicity and Aggregation Properties of RP3-3 and RP3-7

Isolates RWP-3 and RWP-7 had wide range of hydrophobicity features in chloroform and xylene in a time dependent manner (Figure 1B). The percentages of hydrophobicity in chloroform for RWP-3 and RWP-7 were 30.83 ± 2.4% vs. 31.72 ± 1.2% at 30 min, 48.36 ± 1.6% vs. 37.33 ± 3.8% at 90 min, and 70.18 ± 1.1% vs. 62.63 ± 2.1% at 180 min, respectively. Hydrophobicity levels in xylene for RWP-3 and RWP-7were 34.32 ± 1.5 vs. 34.80 ± 2.8%, 35.88 ± 4.1 vs. 37.52 ± 3.7%, and 53.07 ± 5.5 vs. 42.63 ± 4.9%, respectively. In addition, RWP-3 and RWP-7exhibited broad ranges of aggregation ability in a time dependent incubation. RWP3-3 showed higher aggregation ability from 30 min to 180 min compared to RWP-2 (Figure 1C)

Table 1 showed the antibacterial activity of CFS of isolates against different pathogenic bacteria. Both RWP-3 and RWP-7 exhibited the greatest antibacterial activity with inhibitory zones higher than 15 mm against all tested pathogens *E. coli, S. aureus, P. aeruginosa* and *E. faecalis*. The RWP-7 showed the strongest inhibitory activity against *E. coli* (42.0 ± 1.4 mm) compared to RWP-3. But, higher inhibitory zones were noted for *S. aureus* (24.60 ± 2.8), and *P. aeruginosa* (17.8 ± 0.56) for CFS of RWP-3 than the RWP-7.

Both CFS of isolates showed significant antibacterial activity against all tested pathogenic bacteria. Then we analyzed the time and dose-based killing assay of CFS. All pathogenic bacteria were treated with different concentrations of CFS and then growths were determined at every 12 h. The antibacterial activities of CFS have differed in a time and dose-dependent manner. The bacteria growth was increased when the CFS concentrations were decreased with increased incubation times. All the pathogenic bacterial were completely killed at the concentration of 25 mg/mL in all incubation periods (Figure 2A,B).

Further, we determined the minimum inhibitory concentration (MIC) and minimum bactericidal concentration (MBC). The MIC range for RWP-3 was 6.25 to 12.5 mg/mL and for RWP-7 were 6.25 to 25 mg/mL against tested pathogens. MBC ranges for both RWP-3 and RWP-7 were 12.5 to 25 mg/mL. The MIC and MBC were varied between RWP-3 and RWP-7 against tested pathogens (Table 2).

In co-culture, the isolates RWP-3 and RWP-7 showed significant competitive inhibition of pathogens growths compared to mono-culture of pathogens. Each pathogen was co-cultured with each isolate in MRS-NA broth (1:1). Monocultures LAB and pathogens were also inoculated in MRS-NA broth. The growth of LAB and pathogens at different incubation periods on the respective agar media was determined (NA agar for pathogens and MRS agar for LAB growths). The monoculture of pathogens and LABs growths was increased rapidly in respective agar plates. But, the co-culture of pathogens with LAB competitively reduced their growths compared to monocultures. But the growth of the pathogens was reduced drastically when co-cultured with LAB compared to respective monoculture pathogens. Whereas, slight reduction only was noted for LAB growths in MRS agar as compared to the respective monoculture of LABs (Table 3).

## 4. Discussion

Uses of antibiotics are increasingly inefficient to control the growth of the pathogens as resulting in drug resistance spreading worldwide causing it more difficult to treat infections and death. Increases in antibiotic resistance have got great attention to find innovative and alternative measures to treat infectious diseases. Therefore, we focused on the Lactic acid bacteria (LAB) with potent probiotics features and antibacterial activities against various intestinal and urinary tract infections developing pathogens. Nowadays, LABs are considered the most efficient probiotics with potent antimicrobial agents [26,27]. We isolated several LABs from whole crop rice and screened for their antimicrobial activity primarily and selected two LABs based on their microbial activities and identified them at genus and species levels. This data suggested that these strains (RWP-3 and RWP-7) were belonging to *Lactococcus lactis* and could ferment the different carbohydrate substrates and secreted various extracellular enzymes. As potent probiotics, they remain alive during ingestion and in the harsh condition of the gastrointestinal tract, conditions which include the low pH, and bile salts. The survival of ability of *L. lactis* RWP-3 and *L. lactis* RWP-7 in the gastric juice has based on their ability to tolerate acidic pH which is an essential feature of probiotics. *L. lactis* RWP-3 and *L. lactis* RWP-7 incubated in gastric juice for 3h reduced their survival rates at higher than 60%. Even though a reduction in their survival rates, but more than 30% of *L. lactis* RWP-3 and *L. lactis* RWP-7 colonies had received the tolerant property from low pH. The *L. lactis* RWP-3 showed higher tolerance ability in the low pH (gastric juice) than the RWP-7. The physiological concentration of bile salts in the intestinal tract varies between 0.3 to 0.5% [28]. The resistance against bile salts property of probiotic microbes is associated with the bile salt hydrolase activity that reduces the inhibitory activity of bile by hydrolyzing conjugated bile salts [29,30]. The survival ability of RWP-3 and RWP-7 in duodenal juice (pH, 5) significantly improved compared to gastric juice. It suggested that the increased survival abilities of RWP-3 and RWP-7 in the bile salt environment are due to RWP-3 and RWP-7 receiving the resistance capability at the genomic level from the gastric-juice induced stress. Furthermore, these strains were incubated in intestinal juice with pepsin (pH, 8) for another 3h, indicating that the strains were received more tolerant capability from the previous environment compared to duodenal juice. Overall data suggested that the strains were started to get tolerant properties from the stomach and increased their abilities in subsequent unfavorable conditions called the cross-protective stress response [31] and finally reached to intestinal part and homeostasis the gut microbiota and inhibits undesirable microbial growths [32]. Kondrotiene et al. stated as *L. lactis* strains could have survival ability in the low pH (51 and 67%) as well as bile salts (greater than 80%) at various concentrations [33]. But our present study had a slight controversy with previous research; we observed lower survival rates in acidic pH (2.5) for both the isolates whereas higher survival rates in the bile salts were noted for RWP-3 and RWP-7 similar to Kristina et al., report. Bacteria adhesion has been used to assess the adherence ability of probiotics to surface hydrocarbons which is a measure of adhesion to epithelial cells of the GUT [34]. In the present study, we determined the hydrophobicity of isolated strains RWP-3 and RWP-7. The values of hydrophobicity were varied between 30–70% in chloroform and 34–53% in xylene in a time-dependent manner. The highest hydrophobicity values were noted for both strains in the chloroform than the xylene, particularly RWP-3 higher hydrophobicity values in both solvents compared to RWP-7. The hydrophobicity potential is based on organism and strain-specific which can be affected by age and surface chemistry of strains and composition of culture medium [35]. Yerlikaya et al. reported that the *L. lactis* KI showed hydrophobicity values ranging between 3.20 and 89.76%. Our results were significantly concurrent with the results of *L. lactis* KI [36]. Another researcher reported that the nine *L. lactis* strains exhibited hydrophobicity values between 14.9 and 31.3% [37]. Auto-aggregation is an essential factor for biofilm formation that will help with intestinal colonization and bind to intestinal epithelium which controls pathogen adhesion. The isolates RWP-3 and RWP-7 showed significant aggregation properties in a time-dependent manner, confirming that our isolates can interact with mucous or epithelial cells that help to control pathogen adhesion [38]. The antimicrobial activities of lactobacilli are directly associated with the production of organic acids such as lactic acid, acetic acid, propionic acid and other components including hydrogen peroxide, bacteriocins and peptide [39,40]. In the present study, antibacterial effects of isolated strains against *E. coli, S. aureus, P. aeruginosa* and *E. faecalis* were demonstrated by the agar well diffusion, microdilution, MIC/MBC, time-killing assay and co-culture methods. Several, studies have reported that the *Lactobacillus* strains exhibited significant antibacterial activities via the production of metabolites, competition with nutrients utilization, and inhibition of bacterial adhesion to the mucosa and improve the immune response [26,27]. In vitro studies also suggested that *Lactobacillus lactis* exhibited strong antibacterial activities against *E. coli, S. aureus, P. aeruginosa,*
*S. typhimurium* and *E. faecalis* [41,42]. The CFS of RWP-3 and RWP-7 exhibited significant antibacterial activity against tested pathogens. CFS of RWP-3 showed strong antibacterial activity against *E. coli* compared to RWP-3. CFS of RWP-3 showed a higher inhibitory zone against *S. aureus* (*p* < 0.05). There were significant differences among the isolates in their ability to compete with tested pathogens as the rate of inhibition spectrum ranged from 15.6 ± 0.28 to 42.0 ± 1.4 mm. All the tested pathogens had higher sensitivity to the CFS of RWP-3 and RWP-7. Further, concentrations and time-dependent effects of CFS on the pathogenic growths were determined. This data also evidenced that most of the tested pathogens were inhibited at the concentration of 25 mg/mL in all incubatory periods (12–48 h). In addition, other concentrations had significant effects on the growth of pathogens at whole incubation periods compared to control. It suggested that the pathogens growths were inhibited with increased concentration of CFS. This result was concurrent with data obtained from the well diffusion method and time-dependent killing assay. The MIC and MBC of lyophilized CFS of RWP-3 and RWP-7 were determined. This data suggested that the MIC/MBC of both strains have differed among the LAB against different pathogens. RWP-3 showed strong MIC for *E. faecalis* and RWP-7 showed potent MIC for *P. aeruginosa*. The strong MBC was noted for RWP-3 against *E. faecalis* whereas RWP-7 displayed good MBC against *E. coli* and *P. aeruginosa*. Then, we determined the competitive inhibition between LABs and pathogens by the co-culture method. It is a very essential method to find the ability of RWP-3 and RWP-7 to compete for the pathogenic growths because both strains showed significant survival rates in GIT conditions and adherence properties. The co-culture study revealed that the RWP-3 and RWP-7 strongly compete with the growth of pathogens compared to the monoculture of pathogens. Monoculture of both pathogens and isolates were increased their growths in customized media in a time-dependent manner. Co-culture of pathogens with isolates (LAB) strongly reduced the pathogen growths simultaneously RWP-3 and RWP-7 populations were also reduced compared to respective monocultures. However, co-cultured samples showed a drastic reduction in pathogens colonies on Nutrient agar than the RWP-3 and RWP-7 growths on MRS agar. This data suggested that the RWP-3 and RWP-7 strongly compete with the pathogen growth and grow well in co-culture media. Antagonistic activity by different methods evidenced that the RWP-3 and RWP-7 possessed potent inhibitory activity against *E. coli, S. aureus, P. aeruginosa* and *E. faecalis*. Similarly, different isolates of *L. lactis* showed varied antibacterial spectrums against different *E. coli* and other pathogens [36,43].

## 5. Conclusions

In the present study, two *Lactococcus lactis* (RWP-3 and RWP-7) were isolated from whole crop rice and screened for their probiotics potential with antagonistic activity against various infecting causing pathogens. These strains were survived in an acidic environment (pH 2.5), bile salts, and slight alkaline (pH 8) conditions and possessed wide ranges of hydrophobicity properties in chloroform and xylene. In addition, RWP-3 and RWP-7 showed significant auto-aggregation ability in a time-dependent manner, it is believed to be good candidates for potential probiotics. RWP-3 and RWP-7 showed strong antagonistic activities at different ranges against tested pathogens such as *E. coli, S. aureus, P. aeruginosa, and E. faecalis*. In vitro co-culture study revealed that these strains could have the ability to reduce the growth of the pathogens via its competitive inhibition; it suggested that these strains procured might be considered as bio-therapeutic agents instead of antibiotics against bacterial infections. Overall data recommended that the RWP-3 and RWP-7 strains are novel isolates that can be used to control/prevent bacterial infection and improve the gut microbiota of humans and animals.

## Figures and Tables

**Figure 1 microorganisms-09-02513-f001:**
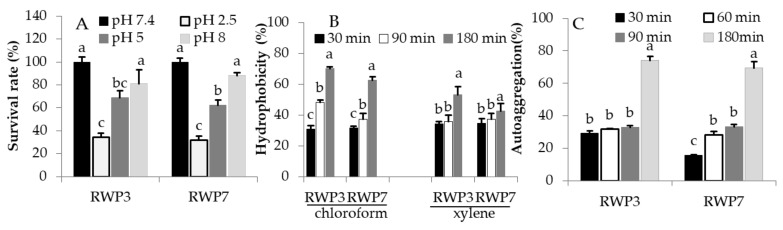
Probiotic characteristics of isolated strains RWP-3 and RWP-7. (**A**) Survival of isolates in GIT conditions, (**B**) Hydrophobicity, (**C**) Auto-aggregation property of isolates. Data were represented as mean ± SEM of three replicates. ^a,b,c^
*p* < 0.05 alphabets between bars represent significance between the pathogens.

**Figure 2 microorganisms-09-02513-f002:**
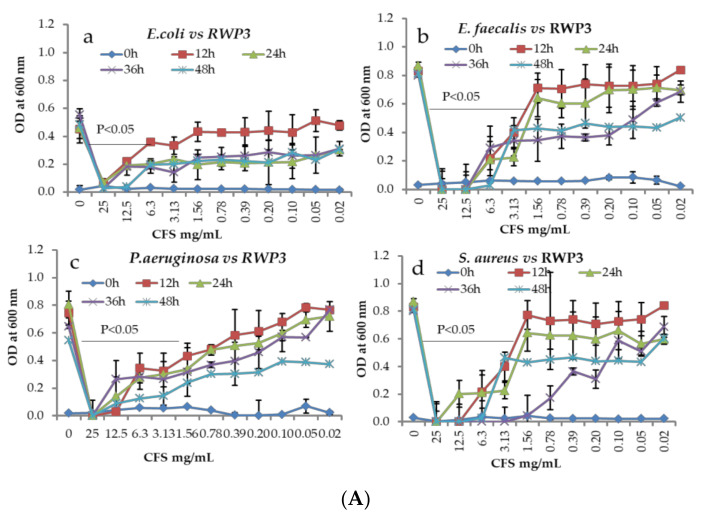
(**A**) Assay of pathogenic bacterial growth inhibition in response to CFS of RWP3 treatment by time and dose-dependent killing method. (**B**) Assay of pathogenic bacterial growth inhibition in response to CFS of RWP7 treatment by time and dose-dependent killing method. (**a**) *E. coli* vs. CFS, (**b**) *E. faecalis* vs. CFS, (**c**) *P. aeruginosa* vs. CFS, (**d**) *S. aureus* vs. CFS. Data were represented as mean ± SEM of three replicates (*n* = 3, *p* < 0.05).

**Table 1 microorganisms-09-02513-t001:** Antibacterial activity by well diffusion method.

Species Name	Zone of Inhibitions (mm)			
	*E. coli*	*S. aureus*	*P. aeruginosa*	*E. faecalis*
RWP-3	23.6 ± 1.2 ^b^	28.60 ± 2.8 ^a^	17.8 ± 0.56 ^c^	18.3 ± 0.49 ^c^
RWP-7	42.0 ± 1.4 ^a^	22.05 ± 2.1 ^b^	15.6 ± 0.28 ^c^	18.5 ± 1.06 ^c^

mm: millimeter. Data were represented as mean ± SEM of three replicates. ^a,b,c^
*p* < 0.05 alphabets within rows represent significance between the pathogens.

**Table 2 microorganisms-09-02513-t002:** Minimum Inhibitory concentration and Minimum bactericidal Concentration (MIC/MBC) of CFS of RWP-3 and RWP-7.

Pathogens	CFS of RWP3		CFS of RWP7	
	MIC (mg/mL)	MBC (mg/mL)	MIC (mg/mL)	MBC (mg/mL)
*E. coli*	12.5	-	25	12.5
*E. faecalis*	6.25	12.5	12.5	25
*P. aeruginosa*	12.5	25	6.25	12.5
*S. aureus*	12.5	25	12.5	25

**Table 3 microorganisms-09-02513-t003:** Growth competition study between *L. lactis* strains and pathogens by co-culture method.

Groups	Growth on MRS Agar	Growth on NA	Groups	Growth on MRS Agar	Growth on NA	Groups	Growth on NA (Mono)
**Bacterial growth (10^7^ CFU/mL) after 12 h**							
**RWP3 alone**	4.28 ± 0.22 ^#^		RWP7 alone	6.08 ± 0.45 ^#^			
**RWP3 ± SA**	3.93 ± 0.15 ^#^	0.236 ± 0.03 ^c^	RWP7 ± SA	5.84 ± 1.20 ^#^	0.70 ± 0.03 ^b^	SA alone	28.0 ± 0.56 ^a^
**RWP3 ± PA**	5.16 ± 0.65 ^#^	0.548 ± 0.06^b^	RWP7 ± PA	6.08 ± 0.11 ^#^	0.50 ± 0.07 ^b^	PA alone	28.4 ± 1.98 ^a^
**RWP3 ± EC**	1.51 ± 0.14 *	0.124 ± 0.02 ^c^	RWP7 ± EC	1.36 ± 0.14 *	0.03 ± 0.01 ^b^	EC alone	29.5 ± 2.80 ^a^
**RWP3 ± EF**	5.06 ± 0.14 ^#^	0.212 ± 0.01 ^c^	RWP7 ± EF	2.97 ± 0.04 ^#^	0.27 ± 0.03 ^c^	EF alone	23.4 ± 0.07 ^a^
**Bacterial growth (10^7^ CFU/mL) after 24 h**							
**RWP3 alone**	5.85 ± 0.11 ^#^		RWP7 alone	7.81 ± 0.30 ^#^			
**RWP3 ± SA**	1.21 ± 0.08 *	0.236 ± 0.03 ^d^	RWP7 ± SA	1.12 ± 0.06 *	0.74 ± 0.03 ^c^	SA alone	54.0 ± 6.29 ^a^
**RWP3 ± PA**	1.26 ± 0.32 *	0.822 ± 0.09^b^	RWP7 ± PA	0.92 ± 0.29 *	0.95 ± 0.28 ^b^	PA alone	47.3 ± 1.81 ^a^
**RWP3 ± EC**	5.50 ± 0.07 ^#^	0.036 ± 0.00 ^e^	RWP7 ± EC	1.30 ± 0.21 *	0.17 ± 0.00 ^d^	EC alone	35.6 ± 2.26 ^a^
**RWP3 ± EF**	1.40 ± 0.14 *	0.680 ± 0.00 ^c^	RWP7 ± EF	1.50 ± 0.78 *	0.17 ± 0.00 ^d^	EF alone	38.7 ± 3.40 ^a^
**Bacterial growth (10^7^ CFU/mL) after 36 h**							
**RWP3 alone**	7.64 ± 0.53 ^#^		RWP7 alone	8.28 ± 0.48 ^#^			
**RWP3 ± SA**	1.68 ± 0.17 *	0.108 ± 0.01^d^	RWP7 ± SA	1.64 ± 0.48 *	0.192 ± 0.01 ^c^	SA alone	48.8 ± 2.95 ^a^
**RWP3 ± PA**	7.44 ± 0.05 ^#^	0.216 ± 0.01 ^c^	RWP7 ± PA	8.14 ± 0.14^#^	0.156 ± 0.03 ^c^	PA alone	33.8 ± 2.07 ^a^
**RWP3 ± EC**	2.80 ± 0.21 *	0.846 ± 0.29 ^b^	RWP7 ± EC	2.71 ± 0.15 *	0.590 ± 0.07 ^b^	EC alone	37.7 ± 2.76 ^a^
**RWP3 ± EF**	3.20 ± 0.28 *	0.16 ± 0.022 ^d^	RWP7 ± EF	7.35 ± 0.81^#^	0.148 ± 0.14 ^c^	EF alone	38.5 ± 5.03 ^a^

Co-Culture: LAB co-cultured with pathogens; Monoculture: LAB and pathogens were cultured separately. Both mono (alone) and co-cultures were grown in MRS: NA broth (1:1 ratio) and incubated at 37 °C with mild shaking. At 12, 24 and 36 h, then tenfold serial dilution as made with sterile distilled water. A hundred microliter of diluted sample was spread onto MRS and NA agar plates for LAB and pathogen growth and incubated at 37 °C for 48 h. The bacterial colonies were counted in both MRS and NA agar plates and the data were compared with respective monocultures. The data were represented as mean ± SEM of three replicates. **,^#^ p* < 0.05 indicates significant between LAB of mono and co-cultures. ^a,b,c,d,e^
*p* < 0.05 alphabets within a columns represent significant between pathogens of mono and co-culture.

## Data Availability

The experimental data are available on request by corresponding author.

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
