# Peer review of "Isolation of Lactococcus lactis from Whole Crop Rice and Determining Its Probiotic and Antimicrobial Properties towards Gastrointestinal Associated Bacteria"

_microorganisms, 2021, doi:10.3390/microorganisms9122513_

Round 1

Reviewer 1 Report

The manuscript "Isolation of Lactococcus Lactis from whole crop rice and determining its probiotic and antimicrobial properties towards gastrointestinal associated bacteria" presents interesting and innovative research results. The manuscript can be published after some minor corrections.

Detailed comments:

Abstract is written correctly and is legible.

Introduction - it is worth adding that LABs can be isolated from silage, e.g. https://doi.org/10.3390/app11177864, doi: 10.1111 / ijfs.14915

Chapter 2.1 - Have the strains been deposited somewhere? Please provide identification methodology, starters, etc.

Chapter 2.2 - please describe the methodology in detail.

The description of the results and the discussion are correct.

Conclusions - more detailed information should be given in this chapter. 

Author Response

We thank the reviewer for giving valuable comments about our research paper which is very helpful to improve the quality of the presentation of the manuscript. We have gone through the whole manuscript according to reviewer suggestions and modified the same. All changes in the manuscript have been made with red color fonts

  1. The manuscript "Isolation of Lactococcus Lactis from whole crop rice and determining its probiotic and antimicrobial properties towards gastrointestinal associated bacteria" presents interesting and innovative research results. The manuscript can be published after some minor corrections.
  2. Abstract is written correctly and is legible.

Thank you for your positive comments on this

  1. Introduction - it is worth adding that LABs can be isolated from silage, e.g.

https://doi.org/10.3390/app11177864, doi: 10.1111 / ijfs.14915

Yes, we agreed with reviewer comments and have been revised as In general, probiotics are represented as a potential alternative for antibiotics to control and prevention of pathogenic bacteria. Strains belonging to lactic acid bacteria such as Lactobacillus and Bifidobacterium are commonly used as a probiotic and starter culture for several fermentation processes (Janiszewska-Turak et al., 2021a; Janiszewska-Turak et al., 2021b; Salminen et al., 2010).

  1. Chapter 2.1 - Have the strains been deposited somewhere? Please provide identification methodology, starters, etc.

Yes, we have deposited these strains to NCBI Gene Bank. The 16srRNA sequence of isolates was analyzed (Solgent Pvt Ltd, Daejeon, Korea) and the sequences were used to identify the isolates at the species level by the NCBI blast tool (GenBank Accession Numbers : RWP-3: OL677065 RWP-7: OL677066. The sequences were aligned with the MUSCLE tool.  The evolutionary history was explored using  the neighbor-joining method(Kumar et al., 2018)

  1. Chapter 2.2 - please describe the methodology in detail.

Yes, now it has been revised as Different ranges of pH (2.5, 5, 8.0 pH values)  in PBS were  prepared with 1M HCl or 1M NaOH and autoclaved at 121°C  with a pressure level at 15psi. The simulated gastrointestinal juices such as gastric juice ( PBS with 3mg/mL pepsin. pH. 2.5) duodenal (PBS with 0.3% bile salts and 0.1% trypsin, pH. 5), and intestinal juice (PBS with 0.1% trypsin, pH.8) were prepared and filtered according to a previously published protocol ((Casarotti and Penna, 2015; Soundharrajan et al., 2019).  Twenty-four bacterial cultures were centrifuged by centrifugation at 4000g for 15 min at 4°C. The collected pellets were washed with PBS and counted by a quantum live-cell staining kit. The pellets were suspended in PBS and the equal numbers of bacterial colonies (1mL) were loaded into 9mL of gastric juice and incubated at 37°C for 3h. One milliliter of bacterial colonies was transferred after 3h from gastric juice to duodenal juice and incubated in the same condition. Similarly, after 3h, 1mL of bacterial colonies were transferred to intestinal juice from duodenal juice and incubated same conditions. The survival of bacterial colonies was determined every one hour. Hydrophobicity features with chloroform and xylene (Kimoto-Nira et al., 2010; Soundharrajan et al., 2019) and aggregation (Del Re et al., 2000; Soundharrajan et al., 2019) properties of strains were also determined.

  1. The description of the results and the discussion are correct.

Thank you for your positive comments on this

  1. Conclusions - more detailed information should be given in this chapter.

According to reviewer suggestion we have revised as in the present study, two lactococcus lactis (RWP-3 and RWP-7) were isolated from whole crop rice and screened for their probiotics potential with antagonistic activity against various infecting causing pathogens. These strains were survived in an acidic environment (pH.2.5), bile salts, and slight alkaline (pH 8) conditions and possessed wide ranges of hydrophobicity properties in chloroform and xylene. In addition, RWP-3 and RWP-7 showed significant auto-aggregation ability in a time-dependent manner, it is believed to be good candidates for potential probiotics. RWP-3 and RWP-7 showed strong antagonistic activities at different ranges against tested pathogens such as E.coli, S. aureus, P. aeruginosa, and E. faecalisInvitro co-culture study revealed that these strains could have the ability to reduce the growth of the pathogens via its competitive inhibition; it suggested that these strains procured might be considered as bio-therapeutic agents instead of antibiotics against bacterial infections. Overall data recommended that the RWP-3 and RWP-7 strains are novel isolates that can be used to control/ prevent bacterial infection and improve the gut microbiota of humans and animals.

Reviewer 2 Report

In the Introduction, the background and the aim should match with logical coherency, the rationale of this study has not been clearly described, specifically, the effect of resistant bacteria in the gut should be described before introducing the possible use of probiotics for antibiotic resistance.

Line 52 should have the proper reference to support that information, line 63 was not clear to me. The acronym should have been introduced after the definition, for example, LAB, GIT, and GRAS were introduced without definition. Alternatively, a separate section for abbreviation at the end might serve the purpose.

Strain identification of Lactococcus lactis should have been presented in SI, maybe a phylogenetic tree would serve the purpose.

It would have been interesting to see if there were any analyses of CFS or any study related to the animal gut system.

It would have been appreciated if there were any comparison between the growth inhibitory activities of  CFS and relevant conventional antibiotics.

Author Response

We thank the reviewer for giving valuable comments about our research paper which is very helpful to improve the quality of the presentation of the manuscript. We have gone through the whole manuscript according to reviewer suggestions and modified the same. All changes in the manuscript have been made with red color fonts

  1. In the Introduction, the background and the aim should match with logical coherency, the rationale of this study has not been clearly described, specifically, and the effect of resistant bacteria in the gut should be described before introducing the possible use of probiotics for antibiotic resistance.

Thank you for your valuable comments on background information. The present study is to explore efficient antibacterial probiotics from natural sources against urinary and gastrointestinal infection-causing pathogens such as E.coli, S. aureus, P. aeruginosa, and E. faecalis. The key aim is to prevent or inhibition of bacterial pathogens, hence we have provided the impact of pathogens on human life globally first in the introduction section with some survey data followed by identification of alternatives for antibiotics such as lactic acid bacteria. We hope the reviewer can understand.

  1. Line 52 should have the proper reference to support that information, line 63 was not clear to me. The acronym should have been introduced after the definition, for example, LAB, GIT, and GRAS were introduced without definition. Alternatively, a separate section for abbreviation at the end might serve the purpose.

Abbreviations for LAB, GIT, and GRAS acronyms have been provided in respective places in the introduction as In addition, enterococcus strains dominate pathogenesis in the gastrointestinal tract (GIT). In general, probiotics are represented as a potential alternative for antibiotics to control and prevention of pathogenic bacteria. Strains belonging to lactic acid bacteria (LAB) such as Lactobacillus and Bifidobacterium are commonly used as a probiotic and starter culture for several fermentation processes. The L. lactis usually synthesize bacteriocins that compounds fight against the pathogenic microbes. The Lactococcus has received the grade of  Generally Recognized as Safe (GRAS) status by the Food and Drug and Administration (Nuryshev et al., 2016).

  1. Strain identification ofLactococcus lactis should have been presented in SI, maybe a phylogenetic tree would serve the purpose.

Yes, we agreed with the reviewer comment and constructed phylogenetic trees for present study sequences and provided them as supplementary figure 1.

  1. It would have been interesting to see if there were any analyses of CFS or any study related to the animal gut system.

Definitely, we accept the reviewers comment. It is an in-vitro study to determine the antibacterial and probiotics efficiency, in the future, surely we will conduct an in-vivo study on small animals (Mouse and rats) and large animals (cow and swine).

  1. It would have been appreciated if there were any comparison between the growth inhibitory activities of CFS and relevant conventional antibiotics.

Yes, we agreed but we have not used any antibiotics as the positive control in this study. We will strictly follow your suggestion in the next study.

Reviewer 3 Report

It is well known that various food products, especially fermented ones, contain potentially probiotic microorganisms. From that point of view, nothing new was discovered. Two new strains have been isolated showing antimicrobial potential. This is shown by these first steps in research, in vitro experiments.
Comments on the paper are:
There are a lot of typos in the paper and care should be taken to writte the names of the bacteria correctly.
Figure 1.
Have statistical analysis of the results been done and can the authors present it in graphs?
Table 2
What was used as a control in this experiment?

Author Response

We thank the reviewer for giving valuable comments about our research paper which is very helpful to improve the quality of the presentation of the manuscript. We have gone through the whole manuscript according to reviewer suggestions and modified the same. All changes in the manuscript have been made with red color fonts

Reviewer 3

  1. It is well known that various food products, especially fermented ones, contain potentially probiotic microorganisms. From that point of view, nothing new was discovered. Two new strains have been isolated showing antimicrobial potential. This is shown by these first steps in research, in vitro experiments.

It is an in-vitro screening for the selection of potent antibacterial probiotics as an alternative for antibiotics. The current data generated by the in-vitro method have suggested that both strains showed significant antibacterial and probiotic potential. In general, it is the first step of discovering antibiotics like activity material. Hereafter, need a lot of optimization steps to enhance secondary metabolites production and to identify the effective molecule and purify the same, it will make the better discovery. In the future we will try to find a target molecule produced by our strains then will try in vivo study.

  1. There are a lot of typos in the paper and care should be taken to write the names of the bacteria correctly.
    Figure 1.

We would like to ask an apology for this careless mistake in the microorganism name and other errors in the manuscript.

  1. Have statistical analysis of the results been done and can the authors present it in graphs?
    Table

Now we have performed statistical analysis generated data using SPSS-16 version software. The statistical significances have been provided in respective tables and figures.

  1. What was used as a control in this experiment?

We have not used any antibiotics as the positive control. We will definitely  follow your suggestion in the next study.